# Analysis of Phenolic Components and Related Biological Activities of 35 Apple (*Malus pumila* Mill.) Cultivars

**DOI:** 10.3390/molecules25184153

**Published:** 2020-09-10

**Authors:** Xiao Liang, Tailin Zhu, Sijia Yang, Xin Li, Bo Song, Yue Wang, Qiong Lin, Jinping Cao

**Affiliations:** 1Laboratory of Fruit Quality Biology/The State Agriculture Ministry Laboratory of Horticultural Plant Growth, Development and Quality Improvement, Zhejiang University, Zijingang Campus, Hangzhou 310058, China; 11816039@zju.edu.cn (X.L.); flannery@zju.edu.cn (T.Z.); 3170100347@zju.edu.cn (S.Y.); Elim1989@163.com (X.L.); 21816133@zju.edu.cn (B.S.); fruit@zju.edu.cn (Y.W.); 2Key Laboratory of Agro-Products Quality and Safety Control in Storage and Transport Process, Ministry of Agriculture and Rural Affairs, Institute of Food Science and Technology, Chinese Academy of Agricultural Sciences, Beijing 100193, China

**Keywords:** apple, phenolics, biological activity, high-resolution mass spectroscopy (HRMS)

## Abstract

Apple (*Malus pumila* Mill.) is a popular fruit with high economic values and various biological activities that are beneficial to human health. In this study, 35 apple cultivars were collected and were evaluated for their basic quality indexes, phenolic compositions, antioxidant activity, anti-tumour, and anti-diabetic activities. The compositions of phenolics were detected by using high-performance liquid chromatography (HPLC) and high-resolution mass spectroscopy (HRMS) assays. The antioxidant activities of peel and pulp extracts from 35 apple cultivars were evaluated by using 1,1-diphenyl-2-picrylhydrazyl (DPPH) scavenging assay and ferric reducing antioxidant power (FRAP) assay. Results showed that the contents of phenolic acids and proanthocyanidins showed significant correlations with the antioxidant activities. Phenolic-rich extracts significantly inhibited HepG2 cell proliferation, with the inhibition activity varied significantly between cultivars. ‘Gold Delicious’ pulp extract, ‘Xiboliyabaidian’ peel and pulp extracts showed protective effects on H_2_O_2_-induced injury of human umbilical vein endothelial cells (HUVEC). ‘Red Fuji’ peel extract, ‘Xiboliyabaidian’ peel and pulp extracts, as well as ‘Gold Delicious’ peel extract, significantly increased glucose consumption of HepG2 cells, in a dose-dependent manner. This research may provide theoretical guidance for further nutritional investigation of the apple resources.

## 1. Introduction

Apple (*Malus pumila* Mill.) is a widely distributed fruit classified into the *Malus* genus of the *Rosaceae* family. It has attracted extensive attention because of its significant economic and nutritional value. Apple fruit is rich in nutritional components, including phenolics, dietary fibres, pectins, and vitamins [1,2,3]. 

As secondary metabolites of plants, phenolics have a broad spectrum of bioactive functions, such as antioxidant activity [4,5,6], anti-tumour [7,8], and anti-inflammatory effects [9], cardiovascular disease prevention [10,11], and weight control [12]. The phenolic components abundant in apples mainly include flavonols, hydroxycinnamic acids, flavan-3-ols, dihydrochalcone, and anthocyanins [13]. The content and distribution of phenolics exhibited various tissue specificity [14]. 

The biologically active components varied among cultivars. Wang et al. [15] compared the antioxidant ability, anticancer ability and flavonoids compositions of 35 citrus cultivars, concluded that phenolics were the chief contributor for the antioxidant ability, and polymethoxylated flavonoids (PMFs) were the dominant contributors of anticancer ability, with the nobiletin as the principle-contributing component of citrus fruit. Ding et al. [16] analysed the bioactive components and antioxidant capacities of peach fruits during two growing seasons, and found that the neochlorogenic acid and chlorogenic acid were the main contributors of antioxidant capacity. Among all cultivars, “Xiang Tao” was deduced as an excellent germplasm because of its rich phenolic acids and high radical scavenging capacity. The comparison of cultivars will help us to explore the main contributing bioactive components, and to discover germplasm with more useful nutritional value. The biologically active components in different apples cultivars have also been analysed, and many cultivars with high content of polyphenols were deduced to be valuable germplasm resources. For example, twenty-two old apple cultivars grown in Poland were found with rich polyphenols, especially phenolic acid, flavan-3-ols, flavonols, and triterpenoids, particularly ursolic acid [17]. Nine old apple cultivars from Tuscany (Italy) showed high polyphenol concentrations (mainly flavanols and phenolic acids) and high total antioxidant capacity [18]. However, most of the relevant research has focused on foreign apple cultivars. As the centre of origin of apple cultivars [19], China is rich in apple germplasm resources, but analysis of phenolic components and related biological activities of these cultivars is still lacking. The evaluation and comparison of apple quality indexes, especially important bioactive substances and functions, are helpful for the exploration and utilization of apple resource as well as the development of apple bioactive substances and the upgrading of industrial chains.

In this study, 35 apple cultivars were collected, and their basic quality indexes were measured. The phenolic components were identified and quantified by high-performance liquid chromatography (HPLC) and high-resolution mass spectroscopy (HRMS). The antioxidant abilities were evaluated by 1,1-diphenyl-2-picrylhydrazyl (DPPH) scavenging assay and ferric reducing antioxidant power (FRAP) assay. HepG2 and human umbilical vein endothelial cell (HUVEC) lines were used to determine the anti-tumour activity, as well as cellular protection and cellular glucose consumption promotion activities. This research aimed to explore the phenolics components and the related biological activities of different apple cultivars in China; thus, to provide a scientific basis for the further study and utilisation of functional nutrients in apple fruits resources.

## 2. Results

### 2.1. Analysis of Basic Indexes of Different Apple Cultivars

Table 1 shows quality indexes of 35 different apple cultivars, which can be classified into three types according to their ripening properties: early maturing cultivars, medium maturing cultivars, and late maturing cultivars. Significant differences were observed in fruit quality indexes, including single fruit weight, colour, firmness, and total soluble solids (TSS). ‘Matsumoto Nishiki’ had the highest fruit weight of 258.19 g, which was 6.8 times of ‘Xiboliyabaidian’ with the lowest fruit weight of 37.74 g. ‘Rizhiwan’ was a red-colour cultivar with the highest color index of red grape (CIRG) value of 2.54, while ‘Tianyisanuowa’, a green-colour cultivar, showed the lowest CIRG value of 1.55. As for firmness, ‘Vista Bella’ showed the lowest firmness value of 0.64 N, while ‘Danguang’ showed the highest firmness value of 42.59 N, which is 66.5 times of the former. ‘Xinshijie’ showed the highest TSS value of 15.99 °Brix while ‘Babusijinuo’ showed the lowest TSS value of 9.63 °Brix. The weight ratio of the peel to the whole fruit varied from 7% to 11% among different cultivar.

### 2.2. Comparison of Total Phenolics, Total Proanthocyanidins, and Antioxidant Activities of Different Apple Cultivars

Table 2 shows total phenolics contents, total proanthocyanidins contents, and antioxidant activities of 35 apple cultivars. The contents of total phenolics and total proanthocyanidins in the peel were significantly higher than those in the pulp. ‘Xiboliyabaidian’ showed the highest total phenolics content (64.76 mg gallic acid equivalents (CAE) /g dry weight of extracts (DW) and total proanthocyanidins content (38.39 mg Procyanidin B2 (PB2) /g DW) in the peel. ‘Gold Delicious’ showed the lowest total phenolics content (12.97 mg CAE/g DW) while ‘Qianqiu’ showed the lowest total proanthocyanidins content (5.42 mg PB2/g DW). ‘Xiboliyabaidian’ also showed the highest total phenolics content (58.11 mg CAE/g DW) and total proanthocyanidins content (38.39 mg PB2/g DW) in the pulp. ‘Xinshijie’ showed the lowest phenolics content (2.39 mg CAE/g DW) while ‘Qianqiu’ showed the lowest proanthocyanidins content (60 mg PB2/g DW).

Table 3 shows that antioxidant activities of apple fruit were significantly correlated with their total phenolics content and total proanthocyanidins content, respectively (R^2^ > 0.91). The antioxidant activity varied significantly among cultivars and tissues, and apple peel extracts showed higher antioxidant activity than pulp extracts. ‘Xiboliyabaidian’ peel showed the highest total phenolics content (64.76 mg CAE/g DW), while ‘Gold Delicious’ peel showed the lowest (12.97 mg CAE/g DW) in all peel samples. ‘Xiboliyabaidian’ pulp showed the highest total phenolics content (58.11 mg CAE/g DW), while ‘Qianqiu’ pulp showed the lowest (2.39 mg CAE/g DW) in all pulp samples. The total phenolic, total proanthocyanidins and antioxidant activities of ‘Xiboliyabaidian’ ranked first among 35 samples, indicating that it was a vital apple germplasm resource deserving the further study.

### 2.3. Comparison of Phenolic Components in Apple Fruit Extracts from Different Cultivars 

Further quantitative analysis and comparison of individual phenolic compounds in 35 different apple cultivars were performed by HPLC and HRMS (Figure 1, Appendix A, Table 4). Eleven chief phenolic compounds were identified from apple fruits combining retention times comparison and spectral characteristics comparison with authorized standards, as well as fragment ions analysis, including cyanidin-3-galactoside, chlorogenic acid, proanthocyanidin B1, proanthocyanidin B2, epicatechin, p-coumaroylquinic acid, quercetin-3-rutinoside, quercetin-3-galactoside, quercetin-3-glucoside, quercetin-3-arabinoside, and phlorizin. They can be classified into anthocyanins, phenolic acids, flavan-3-ols, flavonols, and dihydrochalcones, respectively. Ten of these compounds were calibrated by the standard compounds, while p-coumaroylquinic acid (peak No.6) was relatively quantified with p-coumaric acid. The maximum absorption wavelength, fragment ions information, and retention time of the 11 chief phenolic compounds are shown in Table 4. 

1, Cyanidin-3-galactoside; 2, Procyanidin B1; 3, Chlorogenic acid; 4, Epicatechin; 5, Procyanidin B2; 6, p-coumaric acid; 7, Quercetin-3-rutinoside; 8, Quercetin- 3-galactoside; 9, Quercetin 3-glucoside; 10, phloridzin; 11, Quercetin-3-arabinoside.

The contents of individual phenolics compounds in the peel and pulp of 35 apple cultivars are shown in Figure 2 as a heatmap, and detailed values are shown in Appendix A. Anthocyanins were only found in red apple peel, mainly cyanidin-3-galactoside. ‘Bolan No.8’ showed the highest content of 0.97 mg/g DW, while ‘Mato’ only showed the content of 0.03 mg/g DW. In some green and yellow apples, such as ‘Xiboliyabaidian’, ‘Jingxiang’, and ‘Gold Delicious’, no anthocyanin was detected. Phenolic acids were distributed, in both the pulp and the peel, mainly chlorogenic acid and p-coumaroylquinic acid. The content of chlorogenic acid in the pulp was generally higher than that of the peel. ‘Babusijinuo’ showed the highest content of chlorogenic acid in both peel (2.76 mg/g DW) and pulp (4.39 mg/g DW). Since there was no p-coumaroylquinic acid standard, we quantified p-coumaroylquinic acid relatively with p-coumaric acid. The data in Table 4 indicates that the content of p-coumaroylquinic acid in the pulp was generally larger than that of the peel, and it was not detected in the peel of 13 cultivars such as ‘Mato’, ‘Qianqiu’, and ‘Gold Delicious’. ‘Xiboliyabaidian’ showed the highest content of p-coumaroylquinic acid both in the peel (0.92 mg/g DW) and pulp (0.72 mg/g DW).

The colour change from blue to red in the figure indicates that the substance content is from low to high, with blue indicating the lowest content and red indicating the highest content. The names of apple cultivars were shown as abbreviations: VB for ‘Vista Bella’, MT for ‘Mato’, EM for ‘Early McIntosh’, TYSYW for ‘Tianyisayewa’, CX for ‘Chunxiang’, FX for ‘Faxian’, MN for ‘Matsumoto Nishiki’, BBSJN for ‘Babusijinuo’, XBLYBD for ‘Xiboliyabaidian’, BSM for ‘Basimei’, HL for ‘Honglu’, ND for ‘Nuoda’, PN for ‘Poland No.8’, SS for ‘Sourthern Snap’, QQ for ‘Qianqiu’, GL for ‘Gala’, HMT for ‘Hermhut’, XMF for ‘Ximengfei’, ST for ‘Sakata Tsugaru’, YQ for ‘Yingqiu’, MTH for ‘Mantanghong’, QY for ‘Qiuying’, JX for ‘Jingxiang’, XSJ for ‘Xinshijie’, DG for ‘Danguang’, DN for ‘Dounan’, HH for ‘Huahong’, YH for ‘Yuehong’, NJ for ‘New Jonagold’, RZW for ‘Rizhiwan’, HF for ‘Hanfu’, HG for ‘Huaguan’, RF for ‘Red Fuji’, JG for ‘Jonagold’, GD for ‘Gold Delicious’.

Three flavan-3-ols detected in 35 cultivars mainly included proanthocyanidin B1, proanthocyanidin B2 and epicatechin. Proanthocyanidin B1 in the pulp was richer than that of the peel, and it was not detected in the peel of six cultivars, including ‘Tianyisayewa’, ‘Babusijinuo’, ‘Xiboliyabaidian’, ‘Jingxiang’, ‘Gold Delicious’, and ‘Dounan’. ‘Babusijinuo’ peel showed the highest content of 1.48 mg/g DW, while Jonagold peel showed the lowest content of 0.12 mg/g DW. In the pulp, ‘Xinshijie’ showed the highest content of 3.90 mg/g DW, and ‘Jonagold’ showed the lowest content of 0.04 mg/g DW. The content of epicatechin in the peel was significantly higher than that of the pulp. In the peel, ‘Honglu’ showed the highest content of 3.56 mg/g DW and ‘Gold Delicious’ showed the lowest content of 0.18 mg/g DW, while in the pulp, ‘Xiboliyabaidian’ showed the highest content of 0.89 mg/g DW and ‘Xinshijie’ showed the lowest content of only 0.02 mg/g DW. The content of proanthocyanidin B2 in the peel was also significantly higher than that of the pulp, and nearly half of the apple cultivars were not detected with proanthocyanidin B2 in the pulp. In the peel, ‘Honglu’ also showed the highest content of 3.44 mg/g DW, while ‘Vista Bella’ showed the lowest content of 0.27 mg/g DW. In the pulp, ‘Early McIntosh’ showed the highest content of 1.31 mg/g DW, while ‘Hermhut’ showed the lowest content of 0.12 mg/g DW.

Four flavonols were detected only in the peel of 35 cultivars, all of which were glycosides of quercetin. Among them, quercetin-3-galactoside showed the highest content in the range of 0.25 to 2.57 mg/g DW, followed by quercetin-3-arabinoside (0.05–2.76 mg/g DW) and quercetin-3-glucoside (0.03–0.51 mg/g DW). Quercetin-3-rutinoside showed the lowest content and was not detected in seven cultivars, such as Mato. The content of quercetin-3-rutinoside in the peel of ‘Red Fuji’ was 0.30 mg/g DW, which was the highest, while ‘Sakata Tsugaru’ was the lowest with 0.04 mg/g DW. ‘Honglu’ showed the highest content of both quercetin-3-galactoside and quercetin-3-arabinoside, which was 2.57 mg/g DW and 2.76 mg/g DW, respectively, while ‘Early McIntosh’ showed the highest content of quercetin-3-glucoside of 0.51 mg/g DW. ‘Gold Delicious’ showed the lowest content of all the three substances, which was 0.25 mg/g DW, 0.05 mg/g DW, and 0.03 mg/g DW, respectively. Dihydrochalcone, mainly phloridzin, was detected both in the peel and the pulp of 35 cultivars, and the content in the peel was significantly higher than that of the pulp. It was not detected in the peel of seven cultivars such as ‘Early McIntosh’, and in remaining cultivars, the content ranged from 0.49 to 3.15 mg/g DW. ‘Honglu’ showed the highest content of 3.15 mg/g DW in the peel while ‘Babusijinuo’ showed the highest content of 0.25 mg/g DW in the pulp.

### 2.4. Correlation Analysis of Phenolics Composition and Antioxidant Capacity

In order to analyse the contribution of individual phenolic compounds in the antioxidant activity, the correlation analysis was carried out as shown in Table 5. In the peel, the contents of proanthocyanidin B1, proanthocyanidin B2, epicatechin, and p-coumaric acid showed a significant positive correlation with antioxidant activity. Moreover, in the pulp, the contents of epicatechin and p-coumaroylquinic acid showed a significant positive correlation with antioxidant activity. The results indicate that the antioxidant activity of apple may be mainly contributed by these substances.

### 2.5. Anti-Tumour Proliferation Activity

The in vitro anti-tumour proliferation activity of apple SPE extracts was conducted using human hepatoma liver cancer HepG2 cells (Appendix A). As shown in Figure 3, Figure 4, Figure 5, Figure 6, Figure 7 and Figure 8, when the concentration of SPE extract was lower than 4 μg/mL, none of the groups showed significant inhibitory effect on the growth and proliferation of HepG2 cells, but in the concentration of more than 4 μg/mL, all the groups showed significant inhibitory effects. In some groups, such as ‘Babusijinuo’ peel extract, when the concentration was 100 μg/mL, the inhibitory rate even reached to above 80%. These results showed that the apple extracts had an inhibitory effect on the growth and proliferation of HepG2 cells in a dose-dependent manner.

### 2.6. Correlations between Phenolic Components and Anti-Tumour Activity

The correlation between phenolics components and anti-tumour activity was calculated, as shown in Table 6. Results showed that the anti-tumour activity was correlated with the contents of epicatechin (R^2^ = 0.643) and proanthocyanidin B2 (R^2^ = 0.655) in the peel, while it was correlated with the contents of proanthocyanidin B1 (R^2^ = 0.438), chlorogenic acid (R^2^ = 0.421), epicatechin (R^2^ = 0.536), and proanthocyanidin B2 (R^2^ = 0.417) in the pulp. Most of these phenolic components belong to flavan-3-ols, indicating that flavan-3-ols in apple extracts may play an essential role in inhibiting the growth and proliferation of HepG2.

### 2.7. Protective Effect on HUVEC

Twenty-eight peel and pulp SPE extracts from 14 cultivars (including ‘Red Fuji’, ‘Jonagold’, ‘Gold Delicious’, ‘Xiboliyabaidian’, ‘Honglu’, ‘Early McIntosh’, ‘Babusijinuo’, ‘Basimei’, ‘Jingxiang’, ‘Nuoda’, ‘Rizhiwan’, ‘Mantanghong’, ‘Chunxiang’, ‘Danguang’) were selected to investigate the H_2_O_2_-induced injury protective effect on human umbilical vein endothelial cell lines (HUVEC). Cell viability was measured by SRB assay (Figure 9). Results indicate that the pulp of ‘Gold Delicious’, ‘Xiboliyabaidian’, ‘Babusijinuo’, ‘Jingxiang’, ‘Mantanghong’, and the peel of ‘Xiboliyabaidian’, ‘Basimei’, ‘Jingxiang’, ‘Chunxiang’ showed the significant protective effect on H_2_O_2_-induced injury in HUVEC cells in a dose-dependent manner. After treated by ‘Chunxiang’ and ‘Jingxiang’ peel extract at the maximum concentration, the cell survival rate reached to 56.39% and 53.42%, respectively, nearly twice as much as the negative control (33.63%). Extracts of other cultivars (including ‘Red Fuji’, ‘Jonagold’, ‘Honglu’, ‘Early McIntosh’, ‘Nuoda’, ‘Rizhiwan’, ‘Danguang’) did not show relevant activity (the data are not shown).

### 2.8. Promotion Effect on Glucose Consumption in HepG2 Cells

Seven apple cultivars (including ‘Red Fuji’, ‘Jonagold’, ‘Gold Delicious’, ‘Xiboliyabaidian’, ‘Honglu’, ‘Huaguan’, ‘Chunxiang’) were selected as materials in the glucose consumption experiment. As showed in Figure 10, the addition of 1 mmol/L metformin for 24 h increased the glucose consumption of HepG2 cells by more than 43% compared to the control samples treated with DMSO alone. Among selected cultivars, the peel and pulp extracts from ‘Red Fuji’ and ‘Xiboliyabaidian’ significantly increased glucose consumption with gradient effects. When the concentration of ‘Red Fuji’ peel extract was 0.04, 0.2, and 1 μg/mL, the glucose consumption of HepG2 cells increased by 14.0%, 13.4%, 21.3%, and 41.29%, respectively. When the concentration of ‘Gold Delicious’ peel extract was 0.2, 1, and 5 μg/mL, the glucose consumption increased to 14.6%, 24.3%, 26.3%, and 29.3%, respectively. When the concentration of ‘Xiboliyabaidian’ peel extract was 0.2, 1, 5, and 25 μg/mL, the glucose consumption increased to 14.0%, 18.4%, 24.7%, and 29.3%, respectively, and at the same concentration of pulp extract, the glucose consumption increased to 16.5%, 18.9%, 32.6%, and 45.3%, respectively. Other cultivars (including ‘Jonagold’, ‘Honglu’, ‘Huaguan’, ‘Chunxiang’) extracts did not show activity (the data is not shown).

## 3. Discussion

As a large group of secondary metabolites, polyphenolics are widely found in plants. Apart from the contribution to the colour, flavour, and taste of plants, polyphenolics also have beneficial effects on human health [3]. The apple is an essential source of polyphenolics, with content of 110–357 mg/100g fresh weight (FW) [20,21]. The data have shown that apples are among the top third and fourth dietary sources of total phenolics consumed in America and worldwide, respectively [22,23,24]. Polyphenolic components in apple mainly include phenolic acids, flavonols, flavan-3-ols, anthocyanidins, as well as dihydrochalcones and hydroxycinnamic acids [3,25], when different compounds play essential roles in the total antioxidant activity of apple [26]. It has been reported that the antioxidant activity of apples was highly correlated to the total phenolics content, and epicatechin and procyanidin B2 were the significant contributors to the antioxidant activity of individual compounds [26]. 

In this research, by using DPPH and FRAP assays, we found that the antioxidant activities of apples were significantly correlated with the contents of total phenolics, total proanthocyanidins (including proanthocyanidin B1 and B2), epicatechin, and p-coumaroylquinic acid, which is consistent with the existing reports. Moreover, the total phenolics, total proanthocyanidins and antioxidant contents in the peel were significantly higher than that of the pulp, indicating that most of the phenolic compounds were distributed in the peel rather than the pulp. This also indicates that it is much healthier for us to eat an apple without peeling than with peeling. 

The comparison of phenolic components and antioxidant activities among different apple cultivars has been widely reported, and some cultivars with high polyphenolic contents and antioxidant activities were considered as promising germplasm resources that can be used in commercial cultivation [17,27,28]. As the largest centre of origin and gene diversity of the genus *Malus Mill* [29], China has abundant apple resources, but we still lack systematic research on the bioactive components of these cultivars. In this research, the varieties and contents of phenolic components in the peel and pulp of 35 apple cultivars were analysed comprehensively. Through comparisons, we found that the total phenolics, total proanthocyanidins, and antioxidant activities of ‘Xiboliyabaidian’ ranked first among 35 cultivars. This apple cultivar is native to the former Soviet Union, whose parents are not known. We suppose that it can be selected as an excellent germplasm resource with potential health benefits and commercial values.

A large number of epidemiological investigations have shown that various chronic diseases such as diabetes, cardiovascular diseases, and cancers are related to various environmental factors, dietary habits and lifestyle [30,31,32]. For example, only 5–10% of colon cancer cases are caused by genetic background, and more than 70% of cancers are directly related to the unhealthy diet and sedentary lifestyles [33]. The relationship between fruit and vegetable consumption and human health has been widely studied [3,34,35,36]. Thus, the concept of preventing chronic diseases through lifestyle and dietary changes has been accepted by more and more consumers. Consequently, it is of considerable significance to investigate the use of bioactive components in vegetables and fruits.

Apple extract has anti-mutagenic effects on various mutagenic factors, positively correlated with the content of polyphenolics [37]. For example, after consuming apples, short-chain fatty acids produced by fermentation of phenolic compounds can inhibit histone deacetylase in colon cancer cell lines [38]. Apple extract can also block the NF-κB signalling pathway through the inhibitory effect on oxidative stress, thus reducing the risk of breast cancer [39]. Liver cancer is one of the most lethal forms of cancer with high incidence risk and low survival rate [40]. Thus, in this research, we chose human hepatoma HepG2 cell lines to evaluate anti-tumour activities among different apple cultivars. Results showed that the extracts from peel and pulp of 35 cultivars had significantly inhibitory effects on the proliferation of HepG2 cells. Correlation analysis showed that epicatechin and proanthocyanidin B2 in apple peel had a positive correlation with the anti-tumour activity (R^2^ > 0.6). Similarly, in the pulp, proanthocyanidin B1, chlorogenic acid, epicatechin and proanthocyanidin B2 showed a certain correlation with the anti-tumour activity. It is speculated that these compounds in apple fruit may play major roles in the anti-tumour activity.

The existing studies have shown that the intake of apple polyphenolics in the daily diet can reduce the risk of diabetes [3]. For example, dihydrochalcone in apples, especially phloretin-2-glucoside, inhibits sodium-dependent glucose transportation in the intestine, potentially reducing postprandial blood glucose [41,42], and our research is consistent with these existing reports. We have found that the extracts from ‘Xiboliyabaidian’ peel, ‘Red Fuji’ peel and ‘Xiboliyabaidian’ pulp could significantly increase the glucose consumption of HepG2 cells in a dose-dependent manner. However, the hypoglycaemic activities of other compounds in apple fruit remain to be further explored. Besides, extracts from some apple cultivars also showed significant protective effect on H_2_O_2_-induced injury in HUVEC cells, such as ‘Gold Delicious’ pulp extract and ‘Xiboliyabaidian’ peel, and pulp extract. However, the mechanism is still unclear and needs further investigation.

## 4. Materials and Methods

### 4.1. Materials

Thirty-five apple cultivars at commercial maturity were collected from the national fruit germplasm repository of apple (Xingcheng city of Liaoning Province and Qingdao city of Shandong Province) (Appendix A), and all the cultivars can be classified into three types according to their ripening properties: early maturing cultivars, medium maturing cultivars, and late maturing cultivars. Uniform and injury-free fruits were selected and transported to the laboratory in Zhejiang Province within 24 h. The peel and the pulp tissues were frozen immediately in liquid nitrogen and lyophilized, and then were grounded with a pulveriser and stored at −80 °C. The human hepatoma cell line HepG2 and human umbilical vein endothelial cell line HUVEC were kindly offered by the College of Pharmacy of Zhejiang University.

### 4.2. Measurement of Basic Indexes 

The weight of apple fruit was measured by an analytical balance. The colour index was measured by a MiniScan XE Plus colourimeter (HunterLab, Reston, VA, USA) using the CIE (International Commission on Illumination) 1976 L*a*b* colour difference system measurement standard [43]. Calculate h = arctan b*/a*, C = [(a*)^2^ + (b*)^2^]^0.5^, and calculate CIRG = (180 − h)/(L* + C). Measurement of hardness was carried out using a TA-XT2i Plus texture analyser (Stable Micro System, UK), and the unit was expressed in Newtons (N). The fruit total soluble solids (TSS) was measured using the portable digital display sugar meter PR-101α (ATAGO, Japan), and the unit was expressed using °Brix. The ratio of the peel to the whole fruit = (the whole fruit weight—the whole fruit weight after peeling) / the whole fruit weight × 100%. Ten fruits were used for each cultivar in all the measurements.

### 4.3. Determination of Total Phenolics, Total Proanthocyanidins, and Antioxidant Activity

We extracted 0.25 g of lyophilized peel and pulp powder from 35 apple cultivars using ultrasonication with methanol (containing 1% formic acid) in a material-to-solvent ratio of 1:20 (m/v) for three times, 30 min for each time (ultrasound frequency 60 kHz, power 30 W, JBT/C-YCL500Y/3P, Jining Jinbaite Electronics Co. Ltd,.Jining, China). Then the mixture was centrifuged at 6,000 rpm for 15 min. The supernatants collected from three extractions were used for the determination of total phenolics, total proanthocyanidins and antioxidant activity.

The content of total phenolics was determined by the method of Ghafoor et al. [44] with modifications. In brief, 0.5 mL of appropriately diluted fruit extracts were mixed with 0.5 mL of Folin-Ciocalteu reagent; 1.0 mL 7% Na_2_CO_3_ solution was added to the mixture. After shaking fully, the mixture was put in a water bath at 30 °C for 2 h. Then, the absorbance was measured at 760 nm using an ultraviolet-visible spectrophotometer (UV-2550, Japan), and chlorogenic acid was used as standard. The results were expressed as mg chlorogenic acid equivalent (CAE)/g DW. 

The total proanthocyanidins was determined by the method of Payne et al. [45]. Moreover, 0.03 g of dimethylacetamide (DMAC) was dissolved in 30 mL of ethanol (containing 10% hydrochloric acid) cooled at 4 °C. Then, 50 μL of appropriately diluted fruit extracts were added onto a 96-well plate with three replicates per sample. After adding 250 μL of DMAC solution, the optical delnsity (OD) value of each well was detected by a microplate reader (detection wavelength 640 nm, Thermo Electron Co., Vantaa, Finland). Procyanidin B2 was used as standard, and the results were expressed as mg procyanidin B2 (PB2) equivalent/g DW.

The DPPH assay was carried out according to the method described by Zhang et al. [46]. Moreover, 2 μL of appropriately diluted apple fruit extracts were added onto a 96-well plate, and then 198 μL of 60 μM DPPH solution was added. The mixture was put to react at room temperature for 2 h, using Trolox as a standard. The absorbance at 515 nm was measured by a microplate reader, and the antioxidant activity was expressed as mg Trolox equivalent (TE)/g DW. 

The FRAP assay was carried out according to the method described by Zhang et al. [46]. 100 mL of acetate buffer (300 mmol/L, pH 3.6), 10 mL of TPTZ solution (10 mmol/L TPTZ dissolved in 40 mmol/L HCl), and 100 mL of FeCl_3_ solution (20 mmol/L) was mixed up as FRAP working solution at a ratio of 10:1:10 (v/v/v). Then 100 μL of appropriately diluted apple fruit extracts were mixed with 900 μL of the working solution. After 3–5 min reaction in a light-proof reaction chamber, the absorbance was measured at 593 nm. Trolox was used as a standard, and the antioxidant activity was expressed as mg Trolox equivalent (TE)/g DW.

### 4.4. Analysis of Phenolic Compounds by HPLC and HRMS

We extracted 0.25 g of lyophilised peel and pulp powder from 35 apple cultivars using ultrasonication with methanol (containing 1% formic acid) in a material-to-solvent ratio of 1:20 (m/v) for three times, 30 min (ultrasound frequency 60 kHz, power 30 W). The mixture was centrifuged at 6000 rpm for 15 min, and the supernatants were collected from three extractions. Then, 1.5 mL supernatant of the peel extract and 2.25 mL supernatant of the pulp extract were evaporated on Heidolph rotary evaporator (LABOROTA 4000-efficient, Heidolph Instruments GmbH & CO. KG, Schwabach, Germany) and the residue was dissolved in 150 μL of methanol (containing 1% formic acid). The compounds were filtered through a 0.22 μm membrane and examined using high-performance liquid chromatography (HPLC).

HPLC analysis was carried out according to the method described by Schieber et al. [47] with some modifications. The phenolic compounds were examined with a Waters 2695 liquid phase system (2996 Diode array detector (DAD) detector) coupled with an ODS C18 column (SunFire 5 μL, 4.6 × 250 mm) (Waters, Milford, MA, USA). The mobile phase consisted of 0.1% formic acid (eluent A) and 0.1% formic acid: acetonitrile (50:50, v/v, eluent B) and the injection volume was 10 μL. The gradient program was as follow: 0–45 min, 23–50% of B; 45–65 min, 50–80% of B; 65–68 min, 80–100% of B; 68–73 min, 100% of B; 73–76 min, 100–23% of B; 76–80 min, 23% of B. The detection wavelengths were 280 nm (proanthocyanidins), 330 nm (phenolic acid), 360 nm (flavonol and its glycosides) and 520 nm (anthocyanin). The flow rate was 1 mL/min, and the column temperature was set as 25 °C.

High-resolution mass spectroscopy (HRMS) analysis was performed using AB Sciex Triple TOF 5600+ high-resolution time-of-flight mass spectrometry (Agilent Technologies Inc., Santa Clara, CA, USA). The process was performed by total ion chromatography and electrospray ionisation (ESI). The ion scan range was 100–1500 m/z, and the analysis was operated in negative ion mode ([M − H]^−^). 

### 4.5. Preparation of Apple Fruit SPE Powder

We extracted 10 g of lyophilised apple peel and pulp powder using ultrasonication with acetone (containing 1% formic acid) in a material-to-solvent ratio of 1:20 (m/v) for three times, 30 min for each time in the dark (ultrasonic frequency 60 kHz, power 30 W). The mixture was centrifuged at 8000 rpm for 6–10 min and the supernatants from three separate extractions were combined. The extracts were evaporated under reduced pressure at 35 °C until the acetone was totally removed, and the residue was re-dissolved in aqueous (containing 1% formic acid). The phenolics were enriched by an SPE C18 column (Waters, 12 mL, 2 g) and lyophilized (FM 25EL-85, VirTis, Gardiner, NY, USA) for 24 h to obtain the apple fruit SPE powder for further experience. 

### 4.6. Detection of Anti-Tumour Proliferation Activity

HepG2 cells were cultured in DMEM medium containing 10% inactivated fetal bovine serum (FBS), 2 mmol/L L-glutamine, 100 μg/mL penicillin, and 100 μg/mL of Streptomyces, kept at 37 °C in an incubator (Thermo Scientific 3111, America) containing 5% CO_2_. Cells were replenished with fresh medium every day and subcultured every 2–3 days.

The cell viability was measured using the sulforhodamine B (SRB) assay [48]. Cells in an exponential phase were digested and seeded onto a 96-well plate (5000 cells/well). After being cultured overnight, the appropriate concentration of extracts was added when the control group was incubated with DMSO. The cells were further incubated for three days, and the proliferation rate was measured using SRB according to the manufacturer’s instruction. The survival rate was calculated according to the following formula:Survival rate (%) = OD administration/OD control × 100%(1)

### 4.7. Detection of HUVEC Protection Activity

HUVEC were cultured in DMEM medium containing 10% inactivated fetal bovine serum (FBS), 2 mmol/L L-glutamine, 100 μg/mL penicillin, and 100 μg/mL of Streptomyces, kept at 37 °C in an incubator (Thermo Scientific 3111, America) containing 5% CO_2_. Cells were replenished with fresh medium every day and subcultured every 2–3 days.

The cell viability was measured using the sulforhodamine B (SRB) assay [48]. Cells in an exponential phase were digested and seeded onto a 96-well plate (3,000 cells/well). After being cultured overnight, the appropriate concentration of extracts (5, 10, 20 μg/mL) or NAC was added when the control group was incubated with DMSO. After pre-incubation for 4 h, 0.5 mmol/L H_2_O_2_ was added to the samples. The cells were further incubated for 24 h, and the cell viability was measured using SRB assay according to the manufacturer’s instruction. The survival rate was calculated according to Formula (1). 

### 4.8. Detection of Glucose Consumption Promotion Activity in HepG2 Cells

The glucose consumption experiment was carried out according to the method described by Li et al. [49]. HepG2 cells were incubated in DMEM medium (high glucose) containing 10% FBS. Cells were seeded onto a 96-well plate, and a cell-free well was set as control. When the cells were grown to 80–90% confluence, the medium was removed and washed with PBS buffer, then replaced with serum-free 1640 medium containing 0.2% BSA. DMSO and metformin ((MET), the final concentration of 1 mmol/L) were used as control groups. Different concentrations of the extracts were used in the experiments. After incubating cells for 24 h, the glucose content of each well was determined by the glucose oxidase method. The glucose consumption was calculated according to the following formula:Glucose consumption = glucose content in cell-free blank culture medium − glucose content in culture medium(2)

Three separate experiments were conducted using at least three technical replicates per group. After the glucose content was measured, the cell proliferation rate was measured using SRB according to the manufacturer’s instruction. Finally, the percentage of increase in glucose consumption in each group was assessed compared to the control group.

### 4.9. Statistical Analysis

Statistical analysis was carried out using SPSS version 20.0 (IBM, Armonk, NY, USA). Data were analysed by one-way ANOVA. Multiple comparisons between the groups were performed using the Tukey method. Values were expressed as the mean ± standard deviation.

## 5. Conclusions

The phenolics distribution and bioactivity varied among cultivars and tissues. The peel accumulated more phenolics components than the pulp, which were rich in species and high in contents. Peel extracts contained higher contents of epicatechin, proanthocyanidin B2, and phlorizin than pulp extracts, while pulp extracts contained higher contents of chlorogenic acid, p-coumaroyl quinic acid, and proanthocyanidin B1 than peel extracts. Apple fruit extracts showed inhibitory effect on the proliferation of HepG2 cells, which was correlated with the contents of phenolic acids and flavan-3-ol. Extracts from the ‘Red Fuji’ and ‘Xiboliyabaidian’ peel and ‘Xiboliyabaidian’ pulp significantly increased the glucose consumption of HepG2 cells, indicating the potential hyperglycaemic activity of these apple cultivars. Extracts from the pulp and peel of ‘Xiboliyabaidian’, ‘Basimei’, ‘Jingxiang’, ‘Mantanghong’, ‘Chunxiang’, and ‘Babusijinuo’ alleviated H_2_O_2_-induced injury in HUVEC cells, and significantly increased the cell viability under oxidative damage, indicating high antioxidant activities of apple cultivars. The results of present study provide guidance for the utilisation of apple germplasm resources, especially in terms of biological activity.

## Figures and Tables

**Figure 1 molecules-25-04153-f001:**
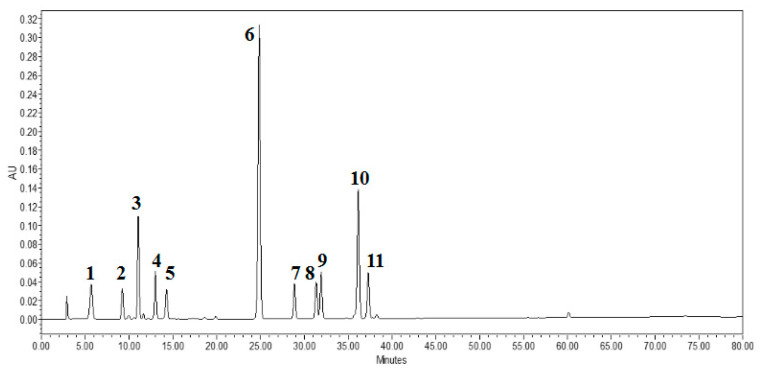
HPLC chromatograms of 11 phenolic compounds in apple fruit (280 nm).

**Figure 2 molecules-25-04153-f002:**
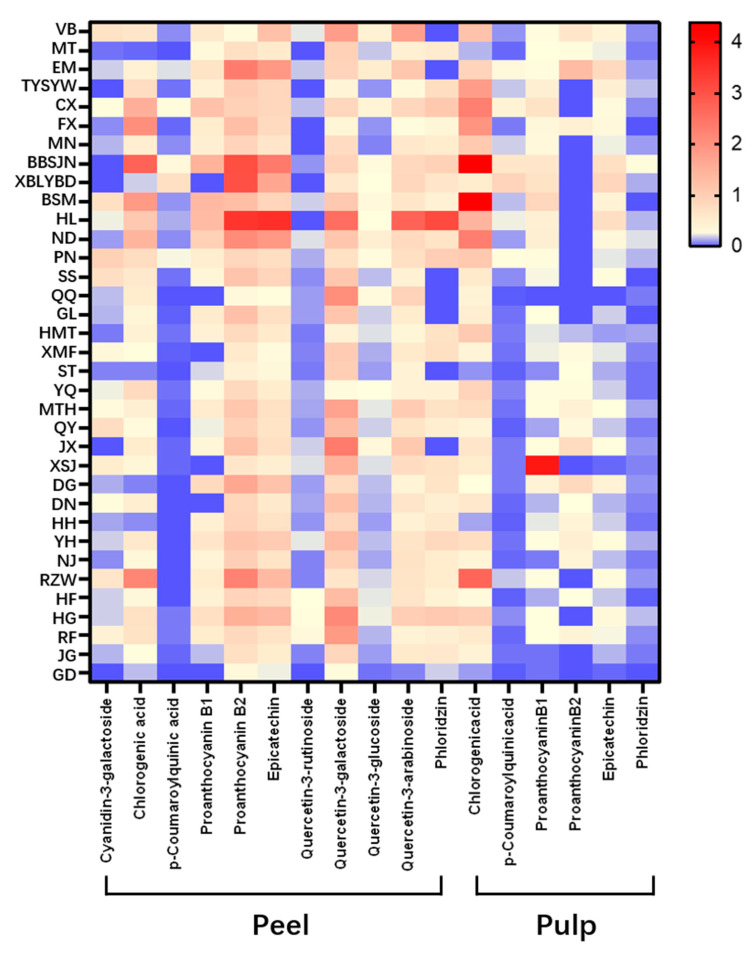
Contents of phenolic compounds in the peel and the pulp of 35 apple cultivars (mg/gdry weight of extracts (DW).

**Figure 3 molecules-25-04153-f003:**
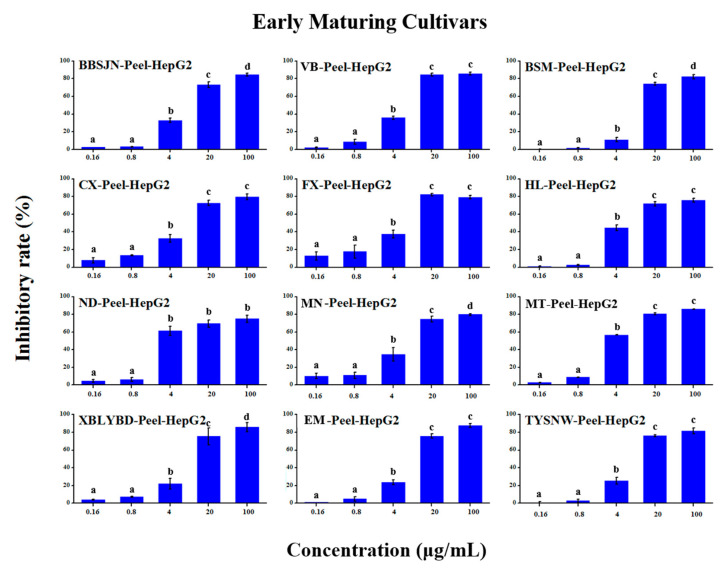
HepG2 growth inhibition effect of peel extracts from early maturing cultivars.The names of apple cultivars were showed as abbreviations: VB for ‘Vista Bella’, MT for ‘Mato’, EM for ‘Early McIntosh’, TYSYW for ‘Tianyisayewa’, CX for ‘Chunxiang’, FX for ‘Faxian’, MN for ‘Matsumoto Nishiki’, BBSJN for ‘Babusijinuo’, XBLYBD for ‘Xiboliyabaidian’, BSM for ‘Basimei’, HL for ‘Honglu’, ND for ‘Nuoda’. Different lowercase letters above the bars indicated significant differences at the 0.05 probability level by Tukey test.

**Figure 4 molecules-25-04153-f004:**
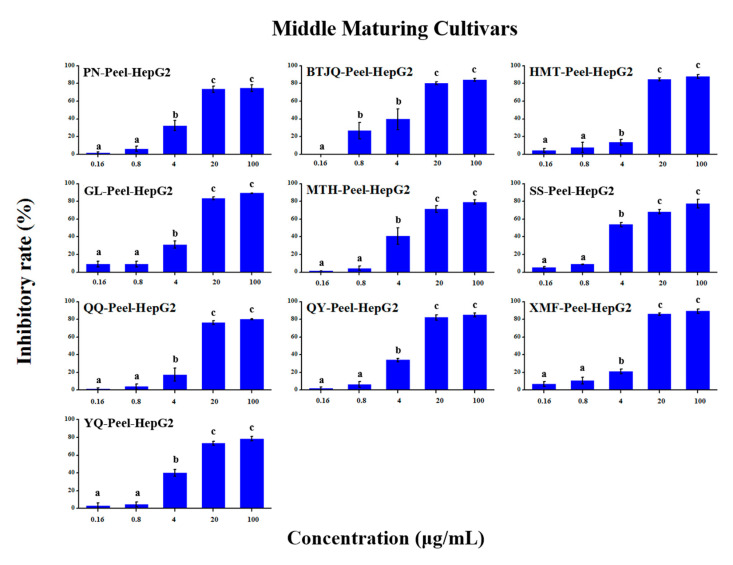
HepG2 growth inhibition effect of peel extracts from middle maturing cultivars.The names of apple cultivars were showed as abbreviations: PN for ‘Poland No.8’, SS for ‘Sourthern Snap’, QQ for ‘Qianqiu’, GL for ‘Gala’, HMT for ‘Hermhut’, XMF for ‘Ximengfei’, BTJQ for ‘ST’, YQ for ‘Yingqiu’, MTH for ‘Mantanghong’, QY for ‘Qiuying’. Different lowercase letters above the bars indicated significant differences at the 0.05 probability level by Tukey test.

**Figure 5 molecules-25-04153-f005:**
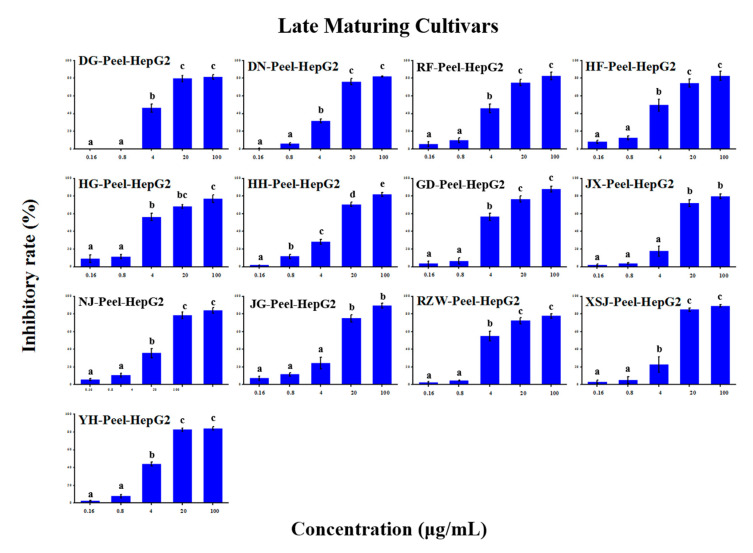
HepG2 growth inhibition effect of peel extracts from late maturing cultivars.The names of apple cultivars were showed as abbreviations: JX for ‘Jingxiang’, XSJ for ‘Xinshijie’, DG for ‘Danguang’, DN for ‘Dounan’, HH for ‘Huahong’, YH for ‘Yuehong’, NJ for ‘New Jonagold’, RZW for ‘Rizhiwan’, HF for ‘Hanfu’, HG for ‘Huaguan’, RF for ‘Red Fuji’, JG for ‘Jonagold’, GD for ‘Gold Delicious’. Different lowercase letters above the bars indicated significant differences at the 0.05 probability level by Tukey test.

**Figure 6 molecules-25-04153-f006:**
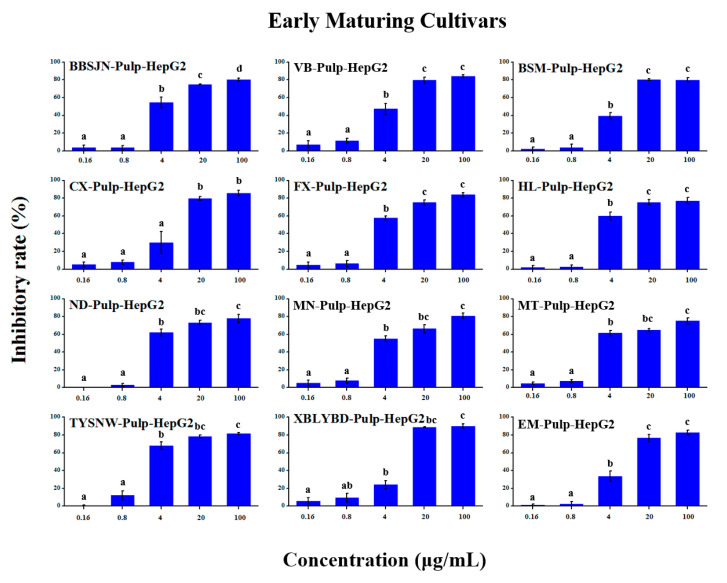
HepG2 growth inhibition effect of pulp extracts from early maturing cultivars.The names of apple cultivars were showed as abbreviations: VB for ‘Vista Bella’, MT for ‘Mato’, EM for ‘Early McIntosh’, TYSYW for ‘Tianyisayewa’, CX for ‘Chunxiang’, FX for ‘Faxian’, MN for ‘Matsumoto Nishiki’, BBSJN for ‘Babusijinuo’, XBLYBD for ‘Xiboliyabaidian’, BSM for ‘Basimei’, HL for ‘Honglu’, ND for ‘Nuoda’. Different lowercase letters above the bars indicated significant differences at the 0.05 probability level by Tukey test.

**Figure 7 molecules-25-04153-f007:**
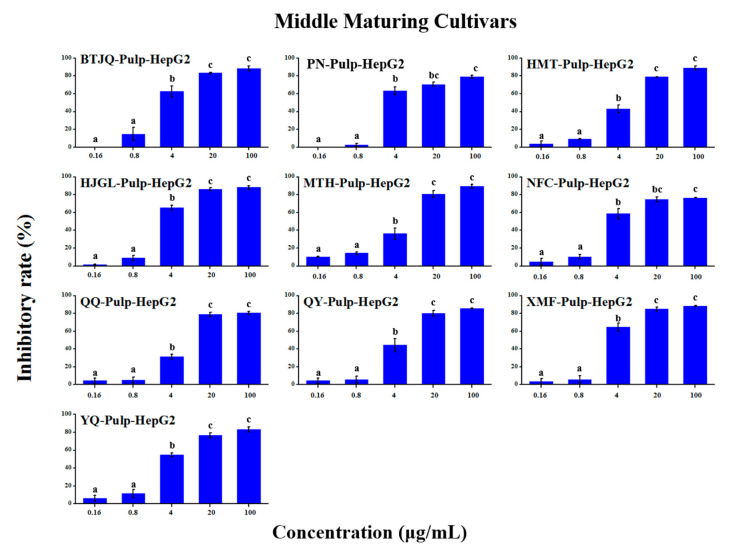
HepG2 growth inhibition effect of pulp extracts from middle maturing cultivars.The names of apple cultivars were showed as abbreviations: PN for ‘Poland No.8’, SS for ‘Sourthern Snap’, QQ for ‘Qianqiu’, GL for ‘Gala’, HMT for ‘Hermhut’, XMF for ‘Ximengfei’, ST for ‘Sakata Tsugaru’, YQ for ‘Yingqiu’, MTH for ‘Mantanghong’, QY for ‘Qiuying’. Different lowercase letters above the bars indicated significant differences at the 0.05 probability level by Tukey test.

**Figure 8 molecules-25-04153-f008:**
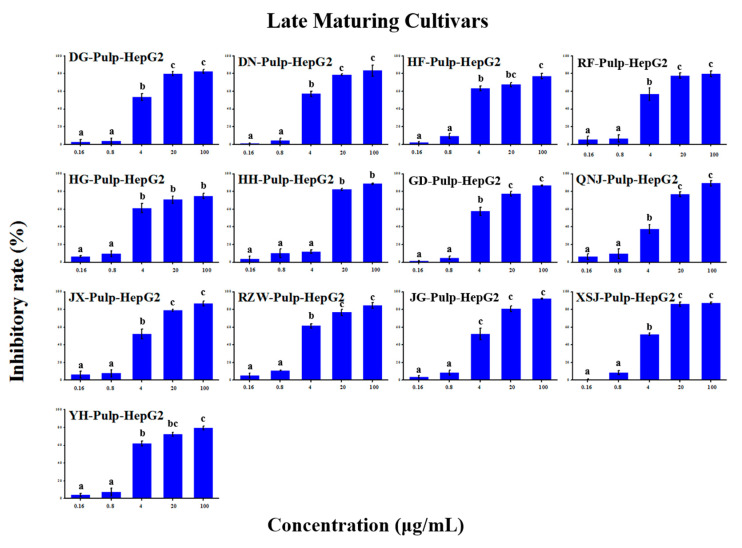
HepG2 growth inhibition effect of pulp extracts from late maturing cultivars.The names of apple cultivars were showed as abbreviations: JX for ‘Jingxiang’, XSJ for ‘Xinshijie’, DG for ‘Danguang’, DN for ‘Dounan’, HH for ‘Huahong’, YH for ‘Yuehong’, NJ for ‘New Jonagold’, RZW for ‘Rizhiwan’, HF for ‘Hanfu’, HG for ‘Huaguan’, RF for ‘Red Fuji’, JG for ‘Jonagold’, GD for ‘Gold Delicious’. Different lowercase letters above the bars indicated significant differences at the 0.05 probability level by Tukey test.

**Figure 9 molecules-25-04153-f009:**
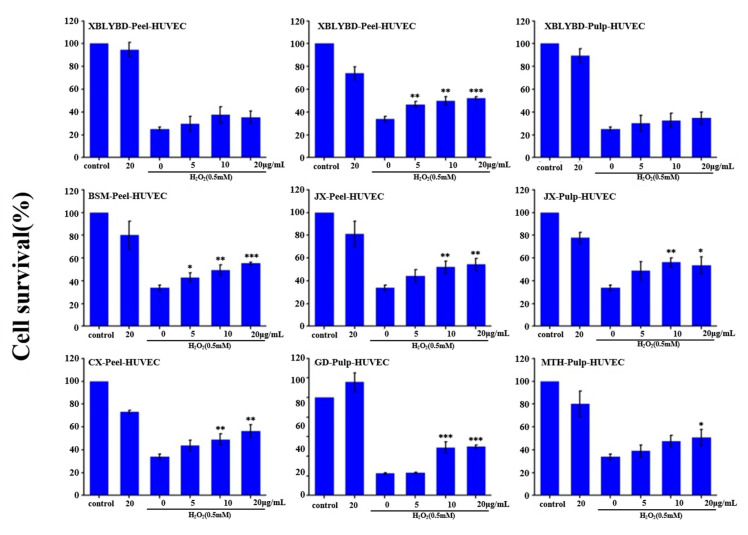
Protective effect of apple extracts on human umbilical vein endothelial cells (HUVEC) under H_2_O_2_-induced stress. The names of apple cultivars were showed as abbreviations: CX for ‘Chunxiang’, BBSJN for ‘Babusijinuo’, XBLYBD for ‘Xiboliyabaidian’, BSM for ‘Basimei’, MTH for ‘Mantanghong’, JX for ‘Jingxiang’, GD for ‘Gold Delicious’. * indicated significant differences at the 0.05 probability level, ** indicated significant differences at the 0.01 probability level, and *** indicated significant differences at the 0.001 probability level compared to the H_2_O_2_ induced group without treating with apple extract using t test.

**Figure 10 molecules-25-04153-f010:**
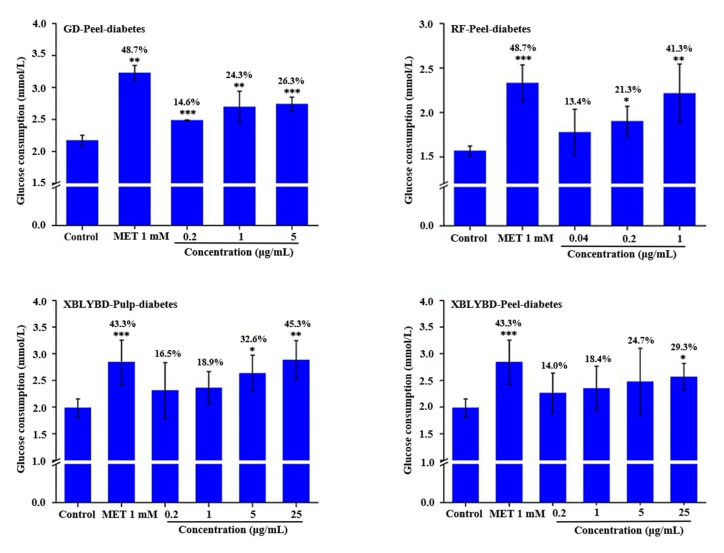
Effect of apple extracts on glucose consumption in HepG2 cells.The names of apple cultivars were showed as abbreviations: XBLYBD for ‘Xiboliyabaidian’, RF for ‘Red Fuji’, GD for ‘Gold Delicious’. * indicated significant differences at the 0.05 probability level, ** indicated significant differences at the 0.01 probability level, and *** indicated significant differences at the 0.001 probability level compared to the control group without treating with apple extract using t test.

**Table 1 molecules-25-04153-t001:** Quality indexes of different apple cultivars.

Mature Period	Cultivar	Fruit Weight (g)	Colour (CIRG)	Firmness (N)	TSS (°Brix)	Ratio of Peel (%)
Early maturing cultivars	Vista Bella	94.43 ± 13.55 ^bcde^	2.18 ± 0.4 ^mnop^	0.64 ± 0.21 ^a^	10.18 ± 0.68 ^ab^	7.99 ± 0.81 ^cdefgh^
	Mato	148.58 ± 9.69 ^hijkl^	1.78 ± 0.17 ^defg^	37.32 ± 5.50 ^lmnop^	10.32 ± 0.46 ^abc^	8.97 ± 1.07 ^defghijk^
	Early McIntosh	82.89 ± 6.06 ^bc^	1.78 ± 0.16 ^defg^	18.52 ± 5.33 ^bc^	10.82 ± 0.74 ^abcde^	11.35 ± 1.19 ^lm^
	Tianyisayewa	119.94 ± 12.99 ^defghi^	1.55 ± 0.04 ^a^	22.99 ± 6.67 ^cd^	10.76 ± 0.80 ^abcde^	8.97 ± 0.70 ^defghijk^
	Chunxiang	95.84 ± 6.51 ^bcdef^	1.91 ± 0.24 ^fghijk^	30.61 ± 3.42 ^fghi^	10.87 ± 0.91 ^abcdef^	8.77 ± 0.77 ^defghij^
	Faxian	119.55 ± 12.53 ^defgh^	1.77 ± 0.12 ^cdefg^	31.16 ± 6.16 ^fghij^	13.03 ± 0.52 ^efghi^	7.79 ± 2.20 ^cdefg^
	Matsumoto Nishiki	258.19 ± 22.59 ^s^	1.79 ± 0.23 ^defgh^	34.89 ± 4.28 ^hijklmn^	11.41 ± 0.75 ^abcdefg^	6.76 ± 1.07 ^abc^
	Babusijinuo	140.93 ± 34.11 ^ghijk^	1.58 ± 0.06 ^abc^	26.71 ± 3.13 ^def^	9.63 ± 0.36 ^a^	8.17 ± 0.74 ^cdefghi^
	Xiboliyabaidian	37.74 ± 7.01 ^a^	1.56 ± 0.04 ^ab^	17.32 ± 2.60 ^b^	/	11.84 ± 0.75 ^m^
	Basimei	76.59 ± 13.01 ^b^	2.05 ± 0.24 ^jklm^	17.90 ± 3.85 ^b^	/	10.38 ± 0.43 ^jklm^
	Honglu	88.16 ± 12.42 ^bcd^	1.75 ± 0.30 ^bcdef^	32.11 ± 2.45 ^ghijk^	13.45 ± 0.56 ^ghi^	8.46 ± 0.50 ^defghij^
	Nuoda	121.37 ± 18.95 ^efghi^	1.76 ± 0.21 ^cdef^	19.70 ± 2.22 ^bc^	10.55 ± 0.64 ^abcd^	8.92 ± 0.83 ^defghijk^
Medium maturing cultivars	Bolan No.8	215.53 ± 30.03 ^qr^	2.20 ± 0.38 ^mnop^	30.42 ± 5.07 ^fgh^	10.87 ± 0.85 ^abcdef^	7.59 ± 0.53 ^cde^
	Southern Snap	116.60 ± 14.72 ^defgh^	2.18 ± 0.23 ^mnop^	34.34 ± 4.61 ^hijklm^	14.51 ± 0.99 ^hij^	9.36 ± 0.92 ^fghijk^
	Qianqiu	164.35 ± 27.75 ^klm^	1.86 ± 0.15 ^efghij^	35.38 ± 3.62 ^ijklmno^	12.38 ± 0.94 ^bcdefghi^	7.97 ± 0.75 ^cdefgh^
	Gala	146.93 ± 22.29 ^hijkl^	1.78 ± 0.14 ^defg^	32.37 ± 5.85 ^ghijk^	12.79 ± 1.18 ^defghi^	8.40 ± 1.13 ^cdefghi^
	Hermhut	127.36 ± 13.47 ^fghij^	1.68 ± 0.16 ^abcde^	31.32 ± 1.60 ^fghij^	10.50 ± 0.57 ^abcd^	9.11 ± 0.49 ^efghijk^
	Ximengfei	112.72 ± 7.03 ^cdefg^	2.37 ± 0.31 ^pqr^	36.36 ± 8.05 ^klmnop^	11.97 ± 1.18 ^abcdefg^	9.54 ± 1.46 ^hijk^
	Sakata Tsugaru	203.54 ± 35.56 ^opqr^	1.81 ± 0.17 ^klmn^	30.67 ± 2.15 ^fghi^	13.08 ± 0.96 ^efghi^	5.76 ± 1.11 ^ab^
	Yingqiu	149.81 ± 22.66 ^ijkl^	1.85 ± 0.20 ^efghi^	25.40 ± 4.20 ^de^	12.54 ± 0.67 ^bcdefghi^	9.59 ± 0.78 ^hijk^
	Mantanghong	142.23 ± 16.17 ^ghijkl^	2.01 ± 0.29 ^ijklm^	39.22 ± 3.58 ^mnopq^	10.97 ± 1.62 ^abcdef^	5.09 ± 0.54 ^a^
	Qiuying	167.06 ± 9.59 ^klmn^	2.44 ± 0.36 ^qr^	40.23 ± 3.60 ^opq^	13.24 ± 0.67 ^fghi^	8.02 ± 0.55 ^cdefgh^
Late maturing cultivars	Red Fuji	219.91 ± 10.98 ^r^	1.82 ± 0.13 ^efghi^	25.47 ± 4.04 ^de^	12.61 ± 1.35 ^cdefghi^	7.62 ± 0.37 ^cde^
	Gold Delicious	211.89 ± 8.07 ^pqr^	1.62 ± 0.03 ^abcd^	19.27 ± 2.42 ^bc^	11.37 ± 0.77 ^abcdefg^	7.58 ± 0.22 ^cde^
	Jonagold	174.01 ± 18.13 ^lmno^	1.97 ± 0.13 ^hijkl^	19.80 ± 2.89 ^bc^	11.34 ± 0.88 ^abcdefg^	7.51 ± 0.27 ^cde^
	Jingxiang	165.73 ± 17.97 ^klm^	1.92 ± 0.13 ^fghijk^	39.55 ± 3.57 ^nopq^	11.54 ± 1.04 ^abcdefg^	10.62 ± 0.46 ^klm^
	Xinshijie	163..95 ± 22.28 ^klm^	2.25 ± 0.22 ^nop^	38.08 ± 3.64 ^mnopq^	15.99 ± 1.16 ^j^	8.42 ± 1.38 ^cdefghi^
	Danguang	182.79 ± 26.60 ^mnop^	2.25 ± 0.23 ^nop^	32.76 ± 3.01 ^ghijkl^	11.59 ± 2.06 ^abcdefg^	9.78 ± 0.89 ^ijkl^
	Dounan	199.18 ± 22.45 ^nopqr^	2.25 ± 0.25 ^nop^	42.59 ± 3.70 ^q^	12.2 ± 0.85 ^bcdefgh^	9.13 ± 0.65 ^efghijk^
	Huahong	159.51 ± 17.02 ^jklm^	1.96 ± 0.16 ^ghijkl^	31.35 ± 2.62 ^fghij^	11.87 ± 1.18 ^abcdefg^	7.37 ± 0.97 ^bcd^
	Yuehong	158.97 ± 15.83 ^jklm^	2.29 ± 0.18 ^opq^	38.53 ± 3.62 ^mnopq^	14.71 ± 1.20 ^ij^	7.97 ± 1.78 ^cdefgh^
	New Jonagold	203.55 ± 16.54 ^opqr^	2.15 ± 0.25 ^lmno^	31.70 ± 3.19 ^ghijk^	12.59 ± 1.05 ^cdefghi^	9.46 ± 0.45 ^ghijk^
	Rizhiwan	183.90 ± 21.05 ^mnopq^	2.54 ± 0.42 ^r^	40.32 ± 6.80 ^pq^	13.00 ± 0.46 ^abcdef^	8.32 ± 1.02 ^cdefghi^
	Hanfu	217.90 ± 34.09 ^r^	2.15 ± 0.23 ^lmno^	28.64 ± 2.87 ^efg^	11.79 ± 0.62 ^a^	7.73 ± 1.07 ^cdef^
	Huaguan	138.71 ± 22.09 ^ghijk^	2.26 ± 0.18 ^nopq^	35.74 ± 3.51 ^jklmnop^	14.26 ± 0.87 ^abcdefg^	7.94 ± 1.48 ^cdefgh^

Different lowercase letters in the same column indicated significant differences at the 0.05 probability level by Tukey test. CIRG = color index of red grape. TSS = total soluble solids.

**Table 2 molecules-25-04153-t002:** Total phenolics, total proanthocyanidins, and antioxidant activities of different apple cultivars.

Cultivars	Total Phenolics (mg CAE/g DW)		Total Proanthocyanidins (mg PB2/g DW)		DPPH (mg TE/g DW)		FRAP (mg TE/g DW)	
	peel	pulp	peel	pulp	peel	pulp	peel	pulp
Vista Bella	18.71 ± 2.17 ^bc^	21.27 ± 1.69 ^q^	20.44 ± 1.77 ^l^	6.07 ± 0.44 ^l^	26.94 ± 2.23 ^lm^	10.59 ± 0.52 ^mno^	43.23 ± 1.57 ^ij^	12.74 ± 1.21 ^klm^
Mato	29.13 ± 4.74 ^hijk^	7.45 ± 0.34 ^defghi^	8.97 ± 0.11 ^bc^	2.80 ± 0.13 ^cd^	18.09 ± 0.27 ^def^	6.51 ± 0.51 ^fgh^	27.97 ± 1.35 ^def^	6.44 ± 0.36 ^cde^
Early McIntosh	54.57 ± 1.79 ^qr^	23.43 ± 1.06 ^r^	28.92 ± 1.21 ^p^	10.69 ± 0.42 ^o^	46.74 ± 0.42 ^n^	21.56 ± 0.46 ^q^	53.56 ± 0.73 ^k^	21.20 ± 1.33 ^o^
Tianyisayewa	26.76 ± 0.91 ^fghi^	12.81 ± 1.14 ^mno^	11.70 ± 0.27 ^de^	5.05 ± 0.22 ^ij^	18.92 ± 0.25 ^defg^	10.78 ± 1.05 ^no^	35.92 ± 1.09 ^g^	14.47 ± 0.77 ^mn^
Chunxiang	38.67 ± 0.51 ^no^	17.29 ± 0.86 ^p^	16.52 ± 1.40 ^ij^	4.42 ± 0.13 ^ghi^	23.58 ± 2.16 ^ghijkl^	8.46 ± 0.62 ^jkl^	38.82 ± 0.85 ^ghi^	13.76 ± 0.30 ^klm^
Faxian	25.49 ± 1.09 ^efgh^	14.04 ± 0.40 ^o^	14.21 ± 0.30 ^fgh^	6.92 ± 0.07 ^m^	22.22 ± 0.54 ^fghijkl^	14.26 ± 0.55 ^p^	27.56 ± 0.80 ^def^	11.94 ± 0.93 ^ijk^
Matsumoto Nishiki	21.86 ± 0.76 ^cde^	9.09 ± 0.61 ^ghijkl^	10.86 ± 0.29 ^cde^	2.80 ± 0.11 ^cd^	20.71 ± 0.81 ^fghi^	6.43 ± 0.22 ^fgh^	30.96 ± 0.69 ^f^	10.32 ± 0.11 ^hi^
Babusijinuo	52.03 ± 1.74 ^pq^	17.40 ± 0.44 ^p^	35.05 ± 0.59 ^r^	8.30 ± 0.35 ^n^	54.31 ± 1.31 ^op^	14.39 ± 0.39 ^p^	64.35 ± 1.49 ^lm^	20.88 ± 0.19 ^o^
Xiboliyabaidian	64.76 ± 1.30 ^s^	58.11 ± 0.67 ^s^	38.39 ± 1.71 ^s^	25.38 ± 0.39 ^p^	68.74 ± 1.28 ^q^	27.80 ± 0.46 ^s^	69.55 ± 3.66 ^n^	40.04 ± 0.80 ^q^
Basimei	48.60 ± 1.15 ^p^	23.43 ± 0.77 ^r^	32.71 ± 0.56 ^q^	10.46 ± 0.42 ^o^	49.56 ± 4.28 ^no^	23.59 ± 1.55 ^r^	61.83 ± 1.13 ^l^	27.28 ± 0.61 ^p^
Honglu	57.73 ± 0.55 ^r^	16.52 ± 0.68 ^p^	36.24 ± 1.12 ^rs^	7.41 ± 0.11 ^m^	55.81 ± 4.88 ^p^	12.04 ± 0.63 ^o^	67.92 ± 1.25 ^mn^	16.15 ± 0.72 ^n^
Nuoda	35.08 ± 0.88 ^mn^	12.07 ± 0.70 ^mn^	24.37 ± 0.48 ^mn^	4.99 ± 0.11 ^hij^	31.58 ± 2.77 ^m^	7.98 ± 0.25 ^hijk^	53.16 ± 0.89 ^k^	14.42 ± 0.52 ^mn^
Bolan No.8	31.05 ± 1.34 ^jkl^	14.32 ± 0.77 ^o^	12.53 ± 0.07 ^efg^	3.36 ± 0.07 ^de^	19.02 ± 0.53 ^defg^	7.30 ± 0.29 ^hij^	43.92 ± 0.86 ^j^	21.72 ± 0.56 ^o^
Southern Snap	24.55 ± 0.68 ^efg^	8.11 ± 0.84 ^efghij^	12.44 ± 0.11 ^ef^	3.51 ± 0.14 ^ef^	20.08 ± 0.61 ^efgh^	6.38 ± 0.17 ^fgh^	41.74 ± 0.83 ^hij^	12.76 ± 0.03 ^jklm^
Qianqiu	15.60 ± 0.84 ^ab^	5.30 ± 0.08 ^bc^	5.42 ± 0.53 ^a^	0.60 ± 0.04 ^a^	12.49 ± 0.10 ^bc^	1.86 ± 0.25 ^a^	18.53 ± 0.45 ^b^	3.28 ± 0.21 ^a^
Gala	24.00 ± 0.70 ^ef^	8.67 ± 0.17 ^fghijk^	15.23 ± 0.17 ^hij^	4.35 ± 0.08 ^gh^	22.72 ± 0.04 ^fghijkl^	6.88 ± 0.50 ^ghij^	25.88 ± 1.66 ^de^	7.35 ± 0.08 ^def^
Hermhut	24.27 ± 0.45 ^efg^	12.78 ± 0.31 ^mno^	10.01 ± 0.37 ^cd^	2.75 ± 0.12 ^cd^	14.12 ± 0.84 ^bcd^	5.59 ± 0.38 ^efg^	40.68 ± 2.66 ^ghij^	14.46 ± 0.24 ^mn^
Ximengfei	22.23 ± 1.29 ^cde^	9.14 ± 0.19 ^hijkl^	6.78 ± 0.15 ^ab^	3.38 ± 0.29 ^de^	12.35 ± 0.44 ^bc^	5.66 ± 0.16 ^efg^	38.27 ± 1.74 ^gh^	12.95 ± 1.91 ^jklm^
Sakata Tsugaru	20.19 ± 0.28 ^cd^	6.15 ± 0.16 ^cd^	7.22 ± 0.09 ^ab^	2.94 ± 0.04 ^cde^	14.07 ± 0.22 ^bcd^	4.75 ± 0.11 ^cde^	20.50 ± 0.58 ^bc^	5.50 ± 0.33 ^bcd^
Yingqiu	23.39 ± 0.47 ^def^	10.07 ± 0.13 ^kl^	10.42 ± 0.10 ^cde^	5.09 ± 0.02 ^jk^	14.37 ± 0.55 ^bcd^	6.61 ± 0.16 ^fghi^	25.83 ± 0.75 ^de^	9.76 ± 0.27 ^gh^
Mantanghong	35.46 ± 1.10 ^mn^	8.28 ± 0.38 ^efghijk^	14.65 ± 0.53 ^ghi^	4.34 ± 0.15 ^gh^	25.31 ± 2.32 ^ijkl^	7.98 ± 0.48 ^hijk^	27.17 ± 0.55 ^def^	12.39 ± 0.11 ^jkl^
Qiuying	29.73 ± 0.51 ^ijk^	7.40 ± 0.12 ^defgh^	11.38 ± 0.27 ^de^	4.14 ± 0.07 ^fg^	15.55 ± 0.43 ^cde^	5.20 ± 0.21 ^def^	27.72 ± 0.95 ^def^	7.29 ± 0.15 ^cdef^
Red Fuji	22.32 ± 1.36 ^cde^	7.26 ± 0.19 ^defg^	14.63 ± 4.28 ^fghi^	2.54 ± 0.14 ^c^	21.48 ± 1.36 ^fghij^	5.31 ± 0.39 ^defg^	29.07 ± 1.71 ^def^	6.84 ± 0.09 ^cdef^
Gold Delicious	12.97 ± 0.26 ^a^	5.14 ± 0.02 ^bc^	/	1.37 ± 0.03 ^b^	10.13 ± 0.17 ^ab^	3.57 ± 0.24 ^bc^	11.80 ± 0.52 ^a^	3.69 ± 0.25 ^ab^
Jonagold	29.53 ± 1.27 ^ijk^	13.93 ± 0.46 ^no^	23.33 ± 0.66 ^m^	5.12 ± 0.21 ^jk^	22.73 ± 2.03 ^fghijkl^	6.84 ± 0.25 ^ghi^	28.26 ± 1.05 ^def^	8.57 ± 0.52 ^fgh^
Jingxiang	41.62 ± 1.17 ^o^	13.67 ± 0.29 ^no^	19.20 ± 0.45 ^kl^	5.73 ± 0.13 ^kl^	21.87 ± 1.01 ^fghijk^	8.21 ± 0.43 ^ijk^	39.34 ± 1.89 ^ghij^	13.69 ± 0.28 ^jklm^
Xinshijie	25.38 ± 0.58 ^efgh^	2.39 ± 0.05 ^a^	6.40 ± 0.11 ^a^	0.61 ± 0.00 ^a^	20.70 ± 0.27 ^fghi^	2.59 ± 0.09 ^ab^	20.78 ± 0.46 ^bc^	7.54 ± 0.35 ^ef^
Danguang	33.85 ± 0.43 ^lm^	10.96 ± 0.78 ^lm^	17.40 ± 0.47 ^jk^	5.85 ± 0.29 ^l^	24.39 ± 0.55 ^hijkl^	9.15 ± 0.30 ^klm^	30.95 ± 1.02 ^f^	4.20 ± 0.24 ^lmn^
Dounan	33.96 ± 1.45 ^lm^	9.60 ± 0.05 ^jkl^	11.78 ± 0.06 ^de^	2.77 ± 0.09 ^cd^	26.86 ± 0.64 ^klm^	6.86 ± 0.08 ^ghij^	28.90 ± 0.22 ^def^	11.84 ± 0.26 ^ijk^
Huahong	27.85 ± 1.10 ^ghij^	9.17 ± 0.21 ^hijkl^	10.12 ± 0.43 ^cd^	3.47 ± 0.08 ^e^	21.60 ± 1.33 ^fghij^	8.19 ± 0.39 ^ijk^	24.60 ± 0.39 ^cd^	6.51 ± 0.48 ^cde^
Yuehong	28.01 ± 1.06 ^ghij^	6.99 ± 0.30 ^cdef^	14.40 ± 0.45 ^fghi^	3.48 ± 0.12 ^e^	26.22 ± 0.99 ^jkl^	9.03 ± 0.50 ^klm^	27.31 ± 1.15 ^def^	8.07 ± 0.04 ^efg^
New Jonagold	26.59 ± 0.29 ^fghi^	8.34 ± 0.05 ^efghijk^	15.97 ± 0.26 ^hij^	3.59 ± 0.28 ^ef^	21.82 ± 1.05 ^fghij^	8.18 ± 0.04 ^ijk^	29.99 ± 0.85 ^ef^	8.52 ± 0.61 ^fgh^
Rizhiwan	39.64 ± 0.96 ^o^	9.30 ± 0.19 ^ijkl^	26.86 ± 0.81 ^op^	4.95 ± 0.10 ^hij^	24.84 ± 1.78 ^hijkl^	9.89 ± 0.42 ^lmn^	36.76 ± 1.74 ^g^	14.26 ± 0.42 ^lmn^
Hanfu	19.12 ± 0.16 ^bc^	3.52 ± 0.05 ^ab^	12.53 ± 0.30 ^efg^	3.21 ± 0.09 ^de^	14.47 ± 0.44 ^bcd^	3.95 ± 0.04 ^bcd^	26.83 ± 0.46 ^def^	5.39 ± 0.28 ^bc^
Huaguan	32.58 ± 0.65 ^klm^	6.63 ± 0.19 ^cde^	26.26 ± 0.59 ^no^	4.29 ± 0.12 ^g^	22.56 ± 0.55 ^a^	6.62 ± 0.41 ^fghi^	36.41 ± 3.45 ^g^	11.78 ± 0.39 ^ij^

Different lowercase letters in the same column indicated significant differences at the 0.05 probability level by Tukey test. GAE = gallic acid equivalents. PB2 = procyanidin B2. TE = trolox equivalent.

**Table 3 molecules-25-04153-t003:** Correlation analysis (R^2^) among total phenolics content, total procyanidins content and antioxidant activities (by DPPH and FRAP assays).

Bioactive Capacities	Total Phenolics	DPPH	FRAP	Total Procyanidins
Total phenolics	1			
DPPH	0.918 **	1		
FRAP	0.906 **	0.914 **	1	
Total procyanidins	0.919 **	0.934 **	0.914 **	1

Note: ** Correlation is significant at 0.01 level.

**Table 4 molecules-25-04153-t004:** Identification of chief phenolic compounds in apple fruit.

Peak No.	RT (min)	λmax (nm)	HRMS/[M−H]^−^	Tentative Identification
1	5.95	279.3, 516.1	/	Cyanidin-3-galactoside (Std)
2	9.22	279.3	/	Procyanidin B1 (Std)
3	11.03	217.8, 325.8	353.0783	Chlorogenic acid (Std)
4	13.0	278.2	/	Epicatechin (Std)
5	14.28	279.3	/	Procyanidin B2 (Std)
6	24.84	226.1, 309.1	337.0930	*p*-Coumaroylquinic acid
7	28.87	255.6, 354.0	609.5102	Quercetin-3-rutinoside (Std)
8	31.13	255.6, 354.1	463.2521	Quercetin-3-glucoside (Std)
9	31.78	255.6, 352.8	463.8649	Quercetin-3-galactoside (Std)
10	36.08	222.5, 284.1	/	Phloridzin (Std)
11	37.49	255.6, 354.0	433.0776	Quercetin-3-arabinoside (Std)

Note: Rt, Retention time; Std: Standard compound, “/”: Data Not Collected.

**Table 5 molecules-25-04153-t005:** Correlation analysis of phenolics content and antioxidant capacity.

Part	Assay	Cyanidin-3-galactoside	Chlorogenic acid	Proanthocyanidin B1	Proanthocyanidin B2	Epicatechin	Quercetin-3-rutinoside	Quercetin-3-galactoside	Quercetin-3-glucoside	Phlorizin	Quercetin-3-arabinoside	*p*-Coumaroylquinic Acid
Peel	DPPH	−0.058	0.384 *	0.524 **	0.840 **	0.781 **	−0.054	0.119	0.422 *	0.387 *	0.515 **	0.729 **
	FRAP	0.116	0.466 **	0.608 **	0.784 **	0.784 **	−0.02	0.105	0.558 **	0.439 **	0.534 **	0.664 **
Pulp	DPPH	/	0.485 **	0.036	0.202	0.837 **	/	/	/	0.221	/	0.707 **
	FRAP	/	0.482 **	0.117	−0.012	0.773 **	/	/	/	0.409 *	/	0.810 **

Note: ** Correlation is significant at 0.01 level, * Correlation is significant at 0.05 level.

**Table 6 molecules-25-04153-t006:** Correlation analysis of phenolic components in apple SPE extract and HepG2 growth inhibition rate.

Bioactive Capacities	Part	Cyanidin-3-galactoside	Procyanidin B1	Chlorogenic Acid	Epicatechin	Procyanidin B2	*p*-Coumaroylquinic Acid	Quercetin-3-rutinoside	Quercetin-3-galactoside	Quercetin-3-glucoside	Quercetin-3-arabinoside	Phloridzin
Inhibition rate #	peel	0.106	0.321	0.274	0.643 **	0.655 **	0.016	0.184	0.106	−0.055	−0.041	/
	pulp	/	0.438 **	0.421 *	0.536 **	0.417 *	−0.004	/	/	/	/	0.128

Note: ** Correlation is significant at 0.01 level; * Correlation is significant at 0.05 level; # HepG2 growth inhibition rate of apple peel extracts at 4 µg/m.

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
