# Peer review of "Analysis of Phenolic Components and Related Biological Activities of 35 Apple (*Malus pumila* Mill.) Cultivars"

_molecules, 2020, doi:10.3390/molecules25184153_

Round 1
Reviewer 1 Report
This research paper, in my opinion, has an average novelty, but it has quite a value for a large number of cultivars (35) compared and quite detailed analysis.
Apple cultivars abbreviations, such as: BL for ‘Vista Bella’, TMYH for ‘Mato’, ZSX for ‘Early McIntosh are quite unclear and confusing for the reader, bus since there is explanation under all tables ant figures, I think it can be leaved as is, because correcting all abbreviations everywhere would be a lot of work. Or maybe authors can explain how these abbreviations are made somewhere in the paper (or comments to reviewers/editors).
English language ant style is very good, easy to read and understand, all sections (introduction, results, etc.) and references are written well (in compliance with all requirements), which shows that this work was written responsibly and carefully.
I suggest accepting this article in it's present form.
Author Response
Comments of Reviewer 1 This research paper, in my opinion, has an average novelty, but it has quite a value for a large number of cultivars (35) compared and quite detailed analysis. Apple cultivars abbreviations, such as: BL for ‘Vista Bella’, TMYH for ‘Mato’, ZSX for ‘Early McIntosh are quite unclear and confusing for the reader, bus since there is explanation under all tables ant figures, I think it can be leaved as is, because correcting all abbreviations everywhere would be a lot of work. Or maybe authors can explain how these abbreviations are made somewhere in the paper (or comments to reviewers/editors). English language ant style is very good, easy to read and understand, all sections (introduction, results, etc.) and references are written well (in compliance with all requirements), which shows that this work was written responsibly and carefully. I suggest accepting this article in it's present form. Reply: We thank the reviewer’s positive comments and helpful suggestions on our manuscript. According to the suggestion, we revised the abbreviations of the cultivars in figure 2-10 to ensure that the abbreviations well corresponded to the full name of the cultivars. (See Fig. 2 – Fig. 10 in the manuscript)Reviewer 2 Report
The research is interesting in the area of the analysis of phenolic components and related biological activities of 35 apple cultivars . Nevertheless, the manuscript needs to be improved in order to provide enough informations to justify its importance and novelty.
Please use the section background to explain the importance and novelty of this research and its objective.
Please conclude according to the main objectives of the research.
Please add previous research on the studied topic.
Author Response
Comments of Reviewer 2 The research is interesting in the area of the analysis of phenolic components and related biological activities of 35 apple cultivars. Nevertheless, the manuscript needs to be improved in order to provide enough informations to justify its importance and novelty. Q1: Please use the section background to explain the importance and novelty of this research and its objective. A1: According to the suggestions, we revised the Introduction and added some information to explain the importance and novelty of the study (See Line 43 - 54). In addition, we added explanation on the objective of this study. (See Line 71 - 72) Q2: Please conclude according to the main objectives of the research. A2: According to the suggestions, we revised the conclusion according to the main objectives of this research. (See Line 522 - 541) Q3: Please add previous research on the studied topic. A3: In the section of Discussion, quite a number of previous studies on this topic had been raised, and full comparison and discussion were done. (See Line 358 - 403 )Reviewer 3 Report
Dear Authors, thank you for your extensive effort for the evaluation of 35 apple caltivars. However, i have some observations.
- As fruits antioxidants property is one of the important biofunctional property so why only 2 methods were used to evaluate, justify please.
- 11 pnenolic compounds were reported as major, HPLC retension time and HRMS chromatogram compare to standard compounds needed to be included as supporting data.
- Please check fig-2, there might be need of correction in the distribution of phenolic compounds between peel and pulp.
- For antitumor proliferation assay please include microscopic image with significance inhibitions.
- Fig-3 to 8 and Fig-8 and 9 , in all cases these are relative small to visualize so there is need to enhancement , beside, some of them might move to the supporting doc.
Author Response
Comments of Reviewer 3 Dear Authors, thank you for your extensive effort for the evaluation of 35 apple caltivars. However, i have some observations. Q1: As fruits antioxidants property is one of the important biofunctional property so why only 2 methods were used to evaluate, justify please. A1: Free radicals scavenging activity and reduction ability were two of the most important aspects of antioxidant abilities of chemical compounds. The most commonly used methods for the free radicals scavenging activity evaluation includes DPPH free radicals scavenging activity evaluation, ABTS free radicals scavenging activity evaluation, oxygen radical absorbance capacity (ORAC) test, etc. The most commonly used method for the reduction ability evaluation was the ferric ion reducing antioxidant power (FRAP) test. In our previous studies on the phenolics antioxidant property evaluation, we had found that the results obtained by evaluation with different antioxidant methods in the same type were highly consistent with each other. Thus, we selected a representative method from each type for the antioxidant property evaluation in present study. Q2: 11 pnenolic compounds were reported as major, HPLC retention time and HRMS chromatogram compare to standard compounds needed to be included as supporting data. A2:According to the suggestion, we added the HPLC retention time and HRMS chromatogram compare to standard compounds as supporting data. (See Figure S1 in the supplementary data). Q3: Please check fig-2, there might be need of correction in the distribution of phenolic compounds between peel and pulp. A3: We made the correction according to the suggestion (See Fig. 2). Q4: For antitumor proliferation assay please include microscopic image with significance inhibitions. A4: According to the suggestion, the microscopic image on the antitumor proliferation assay had been added as supplementary data. (See Fig. S2). Q5: Fig-3 to 8 and Fig-8 and 9, in all cases these are relative small to visualize so there is need to enhancement, beside, some of them might move to the supporting doc. A5: The clarity of the figures had been improved in the revised manuscript according to the suggestion (See Fig. 3 - Fig. 9).Round 2
Reviewer 2 Report
The manuscript has been improved. My recommendation is to be published.